# CODEV-BENCH: HOW DO LLMS UNDERSTAND DEVELOPER-CENTRIC CODE COMPLETION?

## ABSTRACT

Code completion, a key downstream task in code generation, is one of the most frequent and impactful methods for enhancing developer productivity in software development. As intelligent completion tools evolve, we need a robust evaluation benchmark that enables meaningful comparisons between products and guides future advancements. However, existing benchmarks focus more on coarse-grained tasks without industrial analysis resembling general code generation rather than the real-world scenarios developers encounter. Moreover, these benchmarks often rely on costly and time-consuming human annotation, and the standalone test cases fail to leverage minimal tests for maximum repository-level understanding and code coverage. To address these limitations, we first analyze business data from an industrial code completion tool and redefine the evaluation criteria to better align with the developer's intent and desired completion behavior throughout the coding process [1]. Based on these insights, we introduce **Codev-Agent**, an agent-based system that automates repository crawling, constructs execution environments, extracts dynamic calling chains from existing unit tests, and generates new test samples to avoid data leakage, ensuring fair and effective comparisons. Using Codev-Agent, we present the **Co**de-**Dev**elopment Benchmark (**Codev-Bench**), a fine-grained, real-world, repository-level, and developer-centric evaluation framework. Codev-Bench assesses whether a code completion tool can capture a developer's immediate intent and suggest appropriate code across diverse contexts, providing a more realistic benchmark for code completion in modern software development.

## 1 INTRODUCTION

With the rapid development of large language models (LLMs), LLM4X gains popularity cross various domains, including healthcare [8], question answering [23, 24], and education [13]. In software engineering, code LLMs such as Codeqwen [28], Codegeex [37], and Starcoder [20] also demonstrate strong capability in code generation [22, 21]. Several code assistants including TONGYI Lingma [4] and Copilot [17], begin integrating these code LLMs into industrial products. Among these tools, code completion remains one of the most frequent and effective functionalities for boosting developer productivity in daily software development [5]. With the emergence of various tools, there is an urgent need for a comprehensive benchmark to compare their performance [31].

Initially, general code generation benchmarks like HumanEval [4] and ClassEval [6] are used to evaluate completion abilities in form-closed, self-constrained Python functions and classes. However, these benchmarks are too isolated to reflect real-world development practices. In response, DevEval [15] and CrossCodeEval [9] introduce repository-level benchmarks that evaluate the code completion about cross-file contextual understanding and retrieval capabilities, making them more aligned with actual development environments. The latest work, RepoMasterEval [30], further conducts an study to analyze the correlation between user acceptance and benchmark performance, demonstrating that a well-designed benchmark can guide the product improvement in practical settings. However, three key challenges remain: (1) the coarse-grained tasks without industrial analysis resemble general code generation rather than the real-world scenarios developers face; (2) limited extensibility, as

---

[1]A programming tool that suggests and completes code snippets as typing, based on a thorough understanding of the current context of a project. It not just fills in the blanks, but also suggests variable names, functions, classes, methods, and even entire code blocks, depending on the user's desired completion behavior. ——swimm.io

human annotation of data samples and test cases is time-consuming and costly, making continuous updates cumbersome and inflexible; and (3) the standalone generation of test samples fails to leverage minimal tests for maximum repository-level understanding and code coverage.

To address these limitations, we first analyze feedback from an industrial code completion tool and redefine the evaluation criteria to better align with the developer's intent and desired completion behavior throughout the coding process, tackling challenge (1), as discussed in Section 3.1. Based on these insights, we introduce **Codev-Agent** to address challenge (2) and (3), as detailed in Section 3.2. Codev-Agent is an agent-based system that automates repository crawling, constructs execution environments, extracts dynamic call chains from existing unit tests, and generates test samples to avoid data leakage, ensuring fair and effective comparisons. Furthermore, it supports user customization and optimization, allowing users to tailor the benchmark to specific needs and scenarios, making it more adaptable and comprehensive than previous work. With Codev-Agent, we present the developer-centric **Co**de-**Dev**elopment Benchmark (**Codev-Bench**), a fine-grained, real-world, and repository-level evaluation framework, as outlined in Section 3.3. It assesses whether a code completion tool can capture a developer's immediate intent and suggest appropriate code cross diverse contexts, offering a more realistic and actionable evaluation for code completion in modern software development.

After evaluating state-of-the-art general and code-specific LLMs on Codev-Bench, we gain significant insights into their performance and applicability in developer-centric code completion scenarios. Common issues observed include incorrect code indentation, predictions that erroneously span multiple code blocks, incomplete code generation, failure to stop appropriately at the right point, redundant code that repeats the surrounding context, incorrect API calls or parameter values, and misidentification of the correct block type to generate. These problems severely impact the user experience when using code completion tools in practical settings, yet existing benchmarks fail to capture this diverse range of errors. This highlights our core innovation — an automated evaluation framework that assesses code completion from a developer's perspective, ensuring a more comprehensive and realistic evaluation of code completion tools in real-world development environments.

In summary, this paper makes the following contributions to the community:

- We conduct a comprehensive **business analysis** from an industrial code completion product to redefine evaluation criteria that better align with developers' intent and desired completion behavior throughout the coding process.

- We introduce **Codev-Agent**, a unified system that automates the entire process: crawling repositories, setting up execution environments, analyzing call chains from existing unit tests, extracting scenario samples, and evaluating LLMs. Codev-Agent is also adaptable, supporting user customization and optimization to tailor the benchmark to specific needs.

- We propose **Codev-Bench**, a developer-centric, fine-grained, real-world, and repository-level evaluation framework. Codev-Bench assesses whether a code completion tool can capture a developer's immediate intent and suggest appropriate code snippets across diverse contexts, offering a more realistic evaluation for code completion in modern software development.

- We present evaluation results on several state-of-the-art LLMs and provide insights that can guide future research and improvements in the development of LLMs for code completion.

## 2 BACKGROUND AND RELATED WORK

We begin by introducing existing LLMs designed for code generation. Next, we outline the benchmarks for general code generation task and the more specific code completion task. It is important to note that code completion is a key downstream task in code generation, as it is one of the most frequent and impactful methods for enhancing developer productivity in software development. Unlike coarse-grained tasks such as class or function generation, code completion often requires finer granularity. As a result, its evaluation demands more fine-grained and developer-centric scenarios, distinct from those used in general code generation.

### 2.1 LLM FOR CODE GENERATION

The evolution of Code LLMs starts from basic code generation to highly specialized models designed for complex programming tasks. Codeqwen-1.5b [28] marks a step forward with its extensive pretraining on programming languages, refining its precision in code generation. Following this, Deepseek-coder-v2-lite [38] introduces a lightweight, efficient model suited for rapid completion and debugging in constrained environments. The development of Codegeex-4-9b [37] highlights a

shift toward tackling real-world scenarios with its Fill-In-Middle strategy, excelling in mid-segment code generation. Starcoder-2-7b [20] further advances multi-language support, increasing its utility in popular languages like Python and JavaScript. Finally, Codegemma-7b [27] exemplifies the trend toward enterprise-scale code generation, focusing on managing and refining large codebases, meeting the growing needs of software maintenance and development. Together, these models showcase the continuous innovation in Code LLMs, expanding their capabilities and real-world applications.

## 2.2 BENCHMARK FOR CODE GENERATION

Early works introduce benchmarks [35, 10] to evaluate code generation on form-closed and self-constrained Python functions, such as HumanEval [4] and MBPP [3]. ClassEval [6] proposes a class-level code generation dataset containing 100 human-crafted, self-contained Python classes. CoderEval [34] extends the evaluation to non-standalone programs. DevEval [15] and CrossCodeEval [9] align these evaluations with real-world code repositories, addressing more complex code generation scenarios. Meanwhile, XCodeEval [12] and HumanEval-X [37] expand the scope to multilingual programming beyond just Python. In these works, the correctness of generated code snippets, measured by test cases (e.g., Pass@k [34]), and the semantic similarity between generated code and the ground truth (e.g., CodeBLEU [25]) are common evaluation metrics. Test cases derive either from human annotations or from general-purpose LLMs (GPT-4 [1]). The former (human annotation) is time-consuming and costly, while the latter (GPT-4-generated test cases) tends to be unstable and often fails to capture key correlations across a repository.

## 2.3 BENCHMARK FOR CODE COMPLETION

With the emergence of various code assistant tools, such as Copilot [4], Visual Studio IntelliCode [7], and TONGYI Lingma [17], people start to incorporate code LLMs in real-world development. Code completion is one of the most frequent and useful functionalities. CrossCodeEval [5], RepoBench [19], and RepoEval [36] propose benchmarks to evaluate the repository-level code completion across different dimensions, such as cross-file contextual understanding, retrieval capability, and various levels of generation granularity. The most recent work, RepoMasterEval [30], also conducts an industrial study to evaluate the benchmark in a practical setting. However, these work face three common challenges: (1) human annotation of data samples and test cases is time-consuming and costly, (2) coarse-grained tasks such as class and function completion resemble general code generation more than the real-world scenarios developer faced, and (3) unsatisfying and standalone generated test cases fail to reveal code LLMs' ability to understand context at the repository level. To address these gaps, we first analyze the feedback data from industrial code completion tool and redefine the evaluation criteria to better capture a developer's intent and desired completion behavior throughout the coding process. Finally, we deliver the Codev-Agent and Codev-Bench.

## 3 METHODOLOGY

As mentioned earlier, there are three common challenges: (1) coarse-grained tasks resemble general code generation more than real-world scenarios faced by developers, (2) human annotation of data samples and test cases is time-consuming and costly, and (3) standalone generated test cases fail to demonstrate LLMs' ability to understand context at the repository level. To address challenge (1), we request business data from a real-world deployed code completion product through our partner company, which helps us understand the actual needs of developers when using code assistants. For challenges (2) and (3), we introduce an LLM-based agent, Codev-Agent, which automatically crawls up-to-date repositories, constructs execution environments, extracts dynamic call chains from existing unit tests, and generates new test samples based on dynamic data flow to prevent data leakage, ensuring fair and effective comparisons. Built on top of this agent, we provide a fine-grained, real-world, repository-level, and developer-centric benchmark: Codev-Bench.

### 3.1 PRODUCT BUSINESS DATA ANALYSIS

The diverse scenarios in which users trigger code completions involve many variables that affect code assistants' performance. Collaborating with a partner company, we access business data from a real-world code completion product. Analyzing this data helps us understand developers' actual needs and typical usage contexts, guiding the design of a test dataset that mirrors real development environments for more realistic evaluation. Based on the product data, we identify key patterns in how users need code completions:

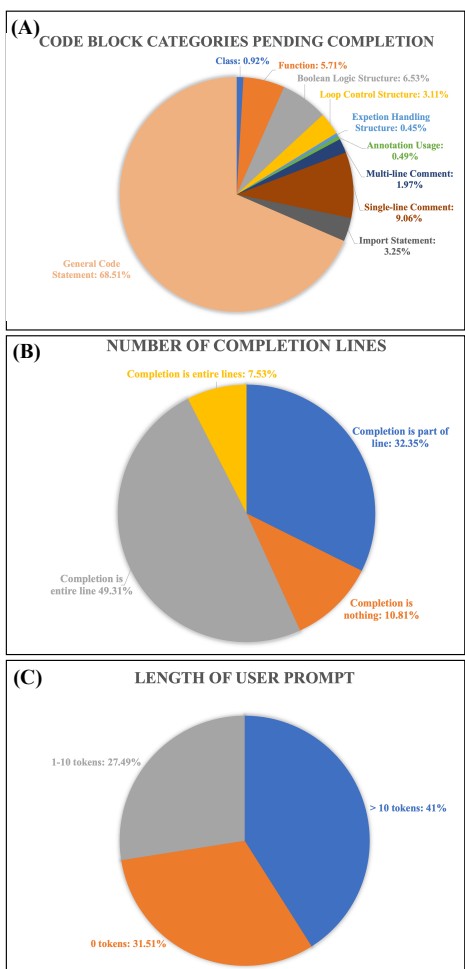

Figure 1: Business data analysis. (A) code block categories distribution. (B) Completion lines distribution. (C) Prompt length distribution.

**Code Block Categories Pending Completion (Figure 1.A)**: The majority of completions, about 68.51%, are triggered for general code statements, reflecting a developer's need for broad, context-aware suggestions. Other significant categories include single-line comments (9.06%) and more specific structures like functions (5.71%) and control logic (6.53%). These distributions highlight the diversity of developer needs and the importance of covering a wide range of code structures in the test dataset.

**Number of Completion Lines (Figure 1.B)**: The data shows that in nearly half of the cases (49.31%), the entire line is completed by the assistant, with a significant portion (32.35%) completing part of a line. This suggests that developers often rely on code assistants for both line completion and generating larger blocks of code, supporting the importance of testing across varying code lengths and completion contexts.

**Length of User Prompts (Figure 1.C)**: We observe that 41% of user prompts contain more than 10 tokens, indicating that developers often provide detailed input to guide code completions. However, there are also cases where prompts contain fewer tokens or even none (31.51%), suggesting that code assistants need to be flexible in handling both detailed and minimal input. This informs our decision to test completions in a variety of prompt conditions, from highly specific to more ambiguous inputs.

By analyzing these key statistics, we ensure that our test dataset aligns closely with real-world usage, covering a broad range of code completion scenarios, varying completion lengths, and differing levels of user input. This data-driven approach enables us to construct a benchmark that accurately reflects the diversity and complexity of developer workflows, ultimately resulting in a more comprehensive and practical evaluation of code completion tools.

## 3.2 CODEV-AGENT

LLM-based Codev-Agent aims to automatically update our benchmark to avoid data leakage, ensuring fair and effective comparisons with minimal cost. It can minimize the human effort but keep the stability in repositories selection, execution environment setup, test samples extraction based on dynamic data flow of existing unit test files. Figure 2 visualizes the pipeline.

**Automated Repository Crawling** As shown in Figure 2a, a LLM-based crawler discovers and crawls up-to-date repositories, gathering the most relevant and current data for analysis and test case generation. The crawler operates based on the following criteria: (1) only repositories created within the last four months are considered to ensure data timeliness; (2) only repositories with a high star count (more than 50 stars) are selected to guarantee community engagement and project popularity; (3) after scanning the file directory, only repositories containing unit test files are retained, ensuring the presence of relevant testing infrastructure; (4) by leveraging the Qwen to understand and analyze the README file, the crawler filters for repositories that require only lightweight configuration environments for execution, thus optimizing for efficiency and reducing complexity.

**Execution Environment Setup** After gathering the qualifying repositories and their associated unit tests from the crawling phase, Codev-Agent sets up the corresponding execution environments through an iterative process. As illustrated in Figure 2b, the first step involves detecting and extracting the unit tests from each repository (Step 0). Next, the agent utilizes Qwen to analyze the README.md

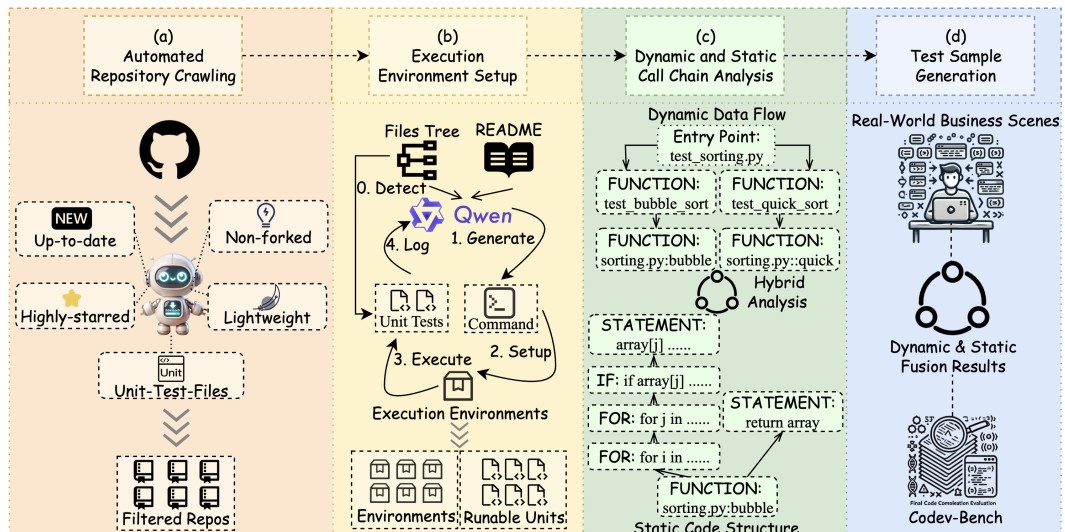

Figure 2: Overview of Codev-Agent. (a) A LLM-based crawler selects up-to-date, lightweight, highly-starred, non-forked repositories with unit test files. (b) Codev-Agent utilize LLM (Qwen) to read README files, generating installation commands and iteratively refining them based on logs and error reports from running unit tests, until successful execution. (c) Codev-Agent combines dynamic data flow analysis during unit test execution with static code parsing (AST), creating a fused code chain that reflects both dynamic and static perspectives. (d) Test Sample Generation extracts test samples from the fusion results based on real-world business scenarios, delivering our Codev-Bench.

file and generate the necessary setup commands for the current environment (Step 1). After the initial setup, the agent executes the extracted unit tests to verify whether they run without errors or warnings (Step 3). Based on the logs from the test executions and command outputs, Qwen determines whether to repeat the process (Steps 1–4), regenerating setup commands if needed, or to finalize the environment setup with executable unit tests.

**Dynamic and Static Call Chain Analysis**  After completing the environment setup and extracting runnable units, Codev-Agent first parses the code files and unit tests in each repository, transforming them into static Abstract Syntax Trees (AST) that include various node types such as Class, Method, If, While, For, Try, Catch, Expression, Statement, and Import, as shown at the bottom of Figure 2c. Simultaneously, Codev-Agent traces the dynamic data flow during the execution of the unit tests, capturing the program's overall behavior and generating the corresponding call chains, depicted at the top of Figure 2c. By fusing the dynamic call chains with the static AST, we can easily extract specific scenarios from the unit test execution process. This module is also adaptable for repositories with few or no existing unit tests, making it an efficient tool for enhancing our benchmark. We can simply design minimal unit tests, and Codev-Agent will automatically and efficiently extract the desired scenarios, saving time spent combing code details through the entire repository.

**Test Sample Generation**  As shown in Figure 2d, once Codev-Agent obtain the dynamic call chains from the unit tests and the static AST structures of the repositories, we can extract the corresponding test samples for each real-world scenario summarized in Section 3.1.

```
{
  "func_name": "function file path and line position",
  "item_dids": ["unit test ids"],
  "unit_test_ids": ["unit test ids"],
  "block_key": "target code block file path and line position",
  "block_type": "AST type of block",
  "prompt": "<filename>xxx<fim_prefix>xxx<fim_suffix>xxx<fim_middle>xxx",
  "prefix": "prefix context of target code block",
  "suffix": "suffix context of target code block",
  "middle": "ground truth of target code block",
  "test_prefix": "prefix context of to construct the unit test",
  "test_suffix": "suffix context of to construct the unit test",
  "test_middle": "ground truth of target code block to construct the unit test",
}
```

Figure 3: Final test sample in JSON format.

For example, if we want to simulate the scenario where a user is writing a function and has just completed the function header but has yet to write the function body, we can start by analyzing a specific unit test. We extract each function that is called or entered during the execution of this unit test. Then, by removing the function body of each extracted function and rerunning the

unit test, if the test becomes unable to execute correctly, we can generate a corresponding prompt based on the current context of the function and save this as a test sample. Following this approach, we systematically extract test samples that match the distribution of real-world business scenarios, using them to simulate realistic code completion use cases based on the business scenarios we summarized. In addtion, Codev-Agent also allows users to design new test units for every repository for augmenting the benchmark with minimal human effort.

**Evaluation Execution**   After completing the benchmark, Codev-Agent prompts each under-evaluate LLM to generate recommended code snippets based on the standard prompt such as Figure 4, and then executes the corresponding unit tests in their respective environments. If the unit tests pass without any errors or warnings using the generated code, it is considered a positive result; otherwise, it is considered negative. Codev-Agent follows this process to test each scenario one by by to conclude a final result.

Figure 4: A sample of prompting under-evaluate LLM to complete. The complete prompt is shown in Appendix C

## 3.3 CODEV-BENCH

With Codev-Agent's support, we deliver a fine-grained, real-world, repository-level, and developer-centric evaluation framework, Codev-Bench. It assesses whether a code completion tool can capture a developer's immediate intent and suggest appropriate code snippets across diverse contexts.

### 3.3.1 FEATURES OF CODEV-BENCH

Table 1: Comparison between existing benchmarks and **Codev-Bench**.

| Benchmark | Extensibility | Auto Annotation | Real Repo | Industry | Granularity | Agent |
|---|---|---|---|---|---|---|
| HumanEval [4] | ✗ | ✗ | ✗ | ✗ | Function | ✗ |
| MBPP [3] | ✗ | ✗ | ✗ | ✗ | Function | ✗ |
| RepoBench [19] | ✗ | ✔ | ✔ | ✗ | Line | ✗ |
| RepoEval [36] | ✗ | ✔ | ✔ | ✗ | Line | ✗ |
| HumanEval+ [18] | ✗ | ✗ | ✗ | ✗ | Function | ✗ |
| ClassEval [6] | ✗ | ✗ | ✗ | ✗ | Class | ✗ |
| CoderEval [34] | ✗ | ✔ | ✔ | ✗ | Function | ✗ |
| EvoCodeBench [14] | ✗ | ✗ | ✔ | ✗ | Function | ✗ |
| CrossCodeEval [5] | ✗ | ✗ | ✔ | ✗ | Line | ✗ |
| RepoMasterEval[30] | ✗ | ✗ | ✔ | ✗ | Line | ✗ |
| **Codev-Bench** | ✔ | ✔ | ✔ | ✔ | Every | ✔ |

**Extensibility:**   Codev-Bench stands out as the most extensible benchmark compared to others. Other baselines rely on manually or LLM-generated datasets where the data samples need to be extracted one by one, and corresponding unit tests must be generated for validation. This process is time-consuming and labor-intensive. In contrast, Codev-Bench uses a dynamic-static analysis module of Codev-Agent as shown in Figure 2.c, which starts from existing unit tests, extracting samples directly from the data flow via simple string-matching operations. This method minimizes effort in validating feasibility. Even for repositories with few or no unit tests, we only need to write the test cases, and the system can generate numerous test data samples from the data flow, making it scalable.

**Auto Annotation:**   Codev-Bench does not rely on LLMs or human intervention for annotation. It extracts annotations directly from the dynamic data flow of the existing unit tests by parsing code. This process ensures that all necessary annotations are derived without any manual effort, unlike other baselines that require manual or LLM-assisted annotations.

**Developer-Centric Benchmark with Industrial Analysis:**   Most prior benchmarks are constructed based on the researchers' understanding of what is needed, without any developer-centric focus. None of the baselines analyze real-world code completion products or utilize real user feedback to identify actual scenarios that developers face. Codev-Bench is the first benchmark to analyze top-tier, industry-level, developer-centric data from an already deployed code assistant tool. As discussed

in Section 3.1, this ensures that Codev-Bench aligns closely with real-world usage. Although RepoMasterEval does touch on some industry analysis, it only explores the relationship between tool's suggestion acceptance and benchmark performance without diving deep into user needs.

**Granularity:** Prior datasets have a coarse, one-size-fits-all granularity, focusing on completing a function body based on its signature, which is far from industrial code completion. Codev-Bench, on the other hand, includes a wide variety of granular, real-world code completion tasks. These range from completing logic blocks like if, for, and while, to completing individual statements, filling in comment sections, completing argument lists for function calls, and more. The diversity and flexibility of these tasks make Codev-Bench much more reflective of actual code completion needs.

**Agent Integration:** Previous benchmarks are based on manually curated datasets, which limits them to small-scale repositories and only provides the final dataset. Codev-Bench, however, offers a fully integrated framework with Codev-Agent, which automates the entire process. This includes crawling repositories, setting up execution environments, analyzing call chains, extracting scenario samples, and evaluating LLMs—all in one unified architecture. It also supports user customization and optimization, allowing them to tailor the benchmarks to specific needs and scenarios, making it far more adaptable and comprehensive than any previous efforts.

In summary, Codev-Bench surpasses existing benchmarks in every key area, offering high extensibility, real-world relevance, comprehensive automation, and fine-grained scenario-based evaluations.

### 3.3.2 BENCHMARK STATISTICS, AND FUTURE EXTENSIONS

Table 2: Statistics for Different Projects

| Metric | Scene1 | Scene2 | Scene3 | Scene4 |
|---|---|---|---|---|
| Ave METHOD | 2.6 | 5.2 | 4.0 | 2.6 |
| Ave IF | 7.7 | 15.4 | 13.9 | 7.7 |
| Ave FOR | 6.3 | 12.6 | 12.6 | 6.3 |
| Ave TRY | 4.6 | 9.2 | 4.6 | 4.6 |
| Ave STATEMENT | 10.0 | 19.6 | 17.9 | 10.0 |

**Statistic** Current statistics of Codev-Bench are in Table 2. For this submission, we select 10 repositories due to storage limitations. These repositories contain a total of 862 code files and 191 existing test cases. After extraction, our test samples cover 55 code files, all 191 test files, and 296 code blocks. In the future, we plan to process additional repositories and make them publicly available on both Hugging Face and GitHub.

**Future Extensions** Codev-Bench currently deliver test samples on Python languages. As our Codev-Agent is adaptable for every language, Codev-Bench can be extended to more languages. In addition, our Codev-Agent allow users to only design new unit tests for every repository for augmenting the benchmark with minimal human effort, further enhancing the extensibility.

## 4 EXPERIMENTS

### 4.1 EXPERIMENTAL SETUP

**Scenario 1 - Full block completion:** In this scenario, the model is tasked with completing a full code block (e.g., function, if, for, try, or statement) based on a complete, unbroken surrounding context. To pass, the model must accurately complete the block and stop at the correct point, ensuring it passes the unit test successfully. **Scenario 2 - Inner block completion:** In this scenario, the model is required to complete a portion of code block based on a complete, unbroken surrounding context. In addition, 20% of the samples in this scenario have an empty ground truth, evaluating the ability to recognize when the current block is already complete and no further completion is needed. **Scenario 3 - Incomplete suffix completion:** Compared to Scenario 1, this scenario focuses on cases where the suffix content following the current cursor is incomplete. It covers two sub-cases: one where all the suffix content after the cursor in entire file is empty, and another where only the content within the current function body after the cursor is missing. **Scenario 4 - RAG-based completion:** In this scenario, the model builds upon the full block completion task by incorporating a Retrieval-Augmented Generation (RAG) module. The repository is partitioned into chunks, with only functions being considered as candidates. The function containing the current code is used as the query, and the query's embedding is compared with the embeddings of the candidate functions. The top 3 most similar candidates are then inserted back into the prompt as hints to guide code generation.

### 4.2 BASE MODELS

**General LLMs.** General LLMs are evaluated in the code understanding. They exhibit diverse strengths in tasks involving complex reasoning and context handling. **Claude-3.5-Sonnet [2]:**

Anthropic designs it for tasks requiring nuanced reasoning, multilingual support, and high-level coding proficiency. It excels in complex multistep workflows, achieving top benchmarks in reasoning (90.4% MMLU) and coding (92% HumanEval). **Deepseek-v2 [16]:** A 236B parameter Mixture-of-Experts model, with only 21B activated per token, reducing training costs by 42.5% and boosting generation throughput by 5.76x. **Mistral-123b [11]:** A 123B parameter model with a 128k context window, supporting multi-languages and 80+ coding languages. It excel in code and reasoning tasks. **Yi-1.5-34b [33]:** An enhanced version of Yi, pre-trained on 500B tokens and fine-tuned on 3M diverse samples, improving in coding, math, reasoning, and instruction-following. **Qwen-2-54b-moe & Qwen-2-72b [32]:** They incorporate mixture-of-experts (MoE), allowing dynamic routing to improve performance while maintaining efficiency. **Llama-3.1-70b & Llama-3.1-405b [29]:** They are iterated with expanding parameter count, resulting in enhanced reasoning and multitask capabilities in long-context scenarios. **GPT4 & 4o [1]:** The most powerful LLM from OpenAI.

**Code LLMs.** We also evaluate specialized code LLMs designed to handle programming tasks. They implement strategies like Fill-In-Middle to enhance coding accuracy and generation. **Codeqwen-1.5b [28]:** Qwen series, tuned specifically for code completion tasks. It leverages extensive pretraining on programming languages to excel in code generation. **Deepseek-coder-v2-lite [38]:** A lightweight version of Deepseek's code models, optimized for rapid completion and debugging in constrained environments. **Codegeex-4-9b [37]:** With its focus on Fill-In-Middle tasks, Codegeex is tailored for real-world code completion scenarios, excelling in solving mid-segment coding problems. **Starcoder-2-7b [20]**: This model builds upon the original Starcoder, aiming to improve multi-language support and efficiency in common code languages like Python and JavaScript. **Codegemma-7b [27]**: A relatively new model, Codegemma focuses on generating and refining large codebases, making it ideal for enterprise-level code maintenance and development.

### 4.3 EVALUATION METRICS

**Test Case Pass Rate:** We generate code snippets by prompting LLMs and inserting them back into the original code, then rerun the corresponding unit tests. If the test passes without errors, the generated completion is considered successful. We use Pass@1 as the metric, which represents the probability that the LLM generates a code snippet that passes on the first attempt. This metric provides a clear evaluation of how effectively the LLM generates valid, executable code completions.

**Edit Similarity (ES)** Levenshtein distance quantifies the number of single-character edits—insertions, substitutions, or deletions—required to transform one sequence of tokens into another [26]. In practice, developers accept approximate code completions and make manual edits. Therefore, ES is a crucial metric for evaluating how closely a generated completion matches the intended result. The Levenshtein distance between two strings is calculated using dynamic programming.

### 4.4 EXPERIMENTAL RESULS

**Full Block Completion in Full Context** We evaluate the performance of general and code-specific LLMs on completing full code blocks based on complete context (Scenario 1) with results in Table 3 and Table 4. General LLMs struggle with function and try blocks, with a Pass@1 of 0.00% for function completion across all general LLMs except GPT-4 and GPT-4o. However, some models perform well on simpler constructs like statements, where Llama-3.1-405b achieves 78.00%. Those code LLMs in fill-in-the-middle mode show much stronger performance. Codegemma leads with a 53.85% on average, significantly outperforming general LLMs across all block types.

*Insight 1: Simpler blocks like statements are easier to complete, while complex blocks (e.g., functions, condition logic blocks, loop logic blocks) remain challenging. Code LLMs outperform general models, but there is still room for improvement, especially in generating accurate and complete function bodies and logical blocks.*

We found that many models struggle to stop generating content at the appropriate position (shown in Appendix D.1). Consequently, even though the models might generate high-quality code, their overall accuracy remains very low—particularly with models like Starcoder-2-7b and Yi-1.5-34b. This is further corroborated by the average lines of generated codes shown in Figure 5. This issue becomes especially pronounced during real user interactions with code completion models. Users not only face longer waiting times for the model's predictions but also need to remove a considerable amount of extraneous code generated by the model, leading to a significant decline in user experience.

**Sensitivity to Internal Block Completion** We evaluate general and code LLMs to complete internal parts of code blocks based on a complete context (Scenario 2) with results in Table 3 and

Table 3: Comparing Pass@1 of different models across multiple scenarios.

| Model | Scenario 1 | Scenario 2 | Scenario 3 | Scenario 4 |
|---|---|---|---|---|
| General LLMs (Natural Language Mode) | | | | |
| GPT-4 | 27.56 | 15.47 | 6.94 | 25.32 |
| GPT-4o | 45.19 | 12.08 | 5.32 | 41.99 |
| GPT-4o-mini | 19.23 | 3.02 | 2.58 | 20.51 |
| Claude-3.5-Sonnet | 17.95 | 48.87 | 7.74 | 16.35 |
| Deepseek-v2 | 3.21 | 24.72 | 1.29 | 8.97 |
| Mistral-123b | 8.65 | 4.15 | 0.48 | 11.86 |
| Yi-1.5-34b | 3.85 | 15.47 | 0.00 | 2.56 |
| Qwen-2-54b-moe | 2.24 | 23.58 | 0.16 | 1.92 |
| Qwen-2-72b | 27.88 | 8.30 | 2.10 | 26.60 |
| Llama-3.1-70b | 23.72 | 5.47 | 1.45 | 25.00 |
| Llama-3.1-405b | 38.78 | 21.89 | 6.77 | 40.38 |
| Code LLMs (Fill-In-Middle Mode) | | | | |
| Codeqwen-1.5 | 39.10 | 54.72 | 5.44 | 40.71 |
| Deepseek-coder-v2-lite | 47.12 | 78.62 | 3.75 | 45.19 |
| Codegeex-4-9b | 16.67 | 32.70 | 0.50 | 17.63 |
| Starcoder-2-7b | 1.28 | 0.00 | 0.25 | 0.96 |
| Codegemma-7b | 53.85 | 77.36 | 4.84 | 55.77 |

\* Due to space limitations, we have omitted the Pass@1 scores for different types of code blocks in each scenario and provided detailed results in the Appendix A.

Table 5. Among general LLMs, performance is inconsistent, with lower scores for try and statement completions. Claude-3.5 performs best with an average Pass@1 of 48.87%, while models like GPT-4o-mini and Mistral underperform, particularly on try and statement blocks. In contrast, code LLMs excel in this scenario, with Codegemma achieving the highest average Pass@1 score of 77.36%, and Deepspeek-coder-v2-lite performing particularly well, scoring 100.00% on function completions.

*Insight 2: Code LLMs significantly outperform general LLMs in internal block completion tasks, particularly on complex code structures. While general LLMs struggle with try and statement blocks, code LLMs show greater sensitivity to internal block context across all block types.*

Many models struggle to effectively recognize whether the current scenario requires code completion within a block (shown in Appendix D.2) or if the existing code snippet is already complete and does not need any additions (shown in Appendix D.3). This subset is designed to assess a model's ability to effectively complete code within a block and to recognize when no completion is necessary. Models that perform poorly in this scenario may cause significant disruption for users in real-world applications of code completion.

**Completion with Incomplete Suffix**   We evaluate completing code blocks when the suffix content following the cursor is incomplete or missing (Scenario 3) in Table 3 and Table 6. General LLMs perform poorly in this task, especially for more complex block types. For example, most general models achieved a 0.00% Pass@1 on function completions, while the highest performing general model, Llama-3.1-405b, only managed 6.77% on average. On the other hand, code LLMs show slightly better performance. Codegemma achieves the highest average Pass@1 of 6.86% across all block types, with a stronger performance on statements and try blocks, scoring 11.94% and 9.77%, respectively. However, the overall performance of both general and code LLMs remains limited in this scenario, indicating the difficulty of completing code accurately when future context is missing.

*Insight 3: Both general and code LLMs fail in incomplete suffix completions, particularly for complex structures such as functions and try blocks. Code LLMs show a slight advantage, but overall performance highlights the challenge of this completion scenario with missing or incomplete suffixes.*

This subset aligns more closely with real user interactions with code completion tools. Our statistics indicate that in over 20% of cases, the cursor position within the function the user is currently writing has no content following it, and sometimes even the entire file has nothing after the cursor position. Therefore, the model's performance in this scenario is crucial for users utilizing code completion tools. We found that both general LLMs and code LLMs do not perform well in this scenario. This is primarily because the contextual information provided by the surrounding code may be insufficient

for completing the current content. This also highlights that the code completion task continues to present significant challenges.

**RAG-Based Completion**   We evaluate the RAG-based completion (Scenario 4) in Table 3 and Table 7.  Among general LLMs, the performance is mixed, with Llama-3.1-405b achieving the highest average Pass@1 score of 40.38%, and notable performance on functions (33.33%) and try blocks (19.57%).  However, most other general LLMs struggle with this task, especially on function completions, achieving a Pass@1 score of 0.00%. Code LLMs, as expected, demonstrate better overall performance. Codegemma leads the group with an average Pass@1 score of 55.77%, showing strong results on statements (70.00%) and try blocks (44.44%). Deepseek-coder-v2-lite also performs well, with an average score of 52.17%, excelling in the retrieval of for and try blocks.

*Insight 4: General LLMs still struggle with RAG-based tasks, particularly for more complex block types, where code LLMs show a clear advantage. Codegemma and Deepspeek can leverage RAG more effectively, achieving significantly better performance across most block types.*

In real project-level code development, referencing code snippets from other files is quite common. The improvements observed in this scenario test set compared to the results from Scenario 1 reflect the model's ability to recognize and utilize similar code snippets from other files for code completion.

Finally, we examine the correlation between the two metrics: test pass rate (Pass@1) and edit similarity (ES). By calculating the results of each model across four scenarios for these two metrics, we found that, apart from Scenario 2, which has a correlation of 0.991, the correlation in the other three scenarios is below 0.85, specifically 0.793 from Scenario 1, 0.529 from Scenario 3, and 0.822 from Scenario 4. The detailed results are presented in the Appendix B. This indicates that, in most cases, using edit similarity as a statistical measure does not objectively reflect the quality of the model's generation in the code completion task. We also show some

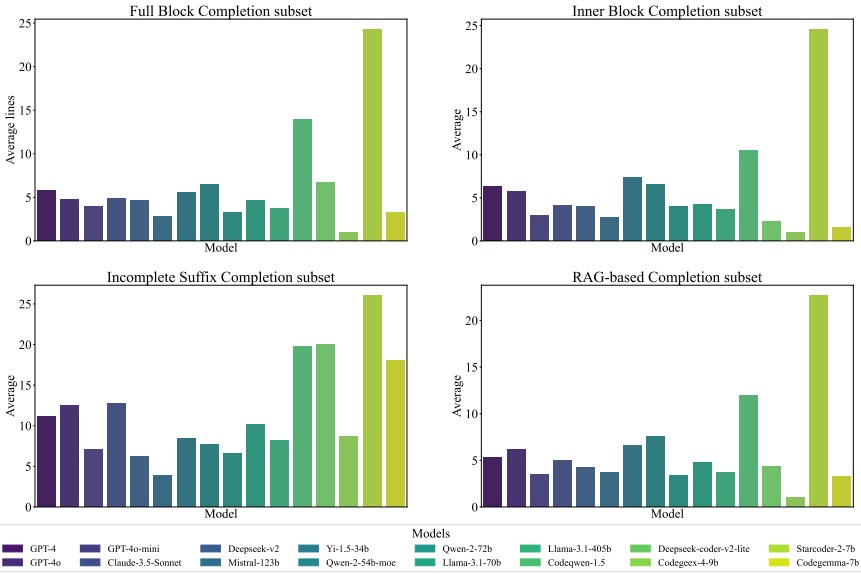

Figure 5: Comparing average lines of generated code by different models in four scenarios.

## 5   SUMMARY AND FUTURE WORK

In this work, we redefined evaluation criteria for code completion tools through a comprehensive business analysis and introduced Codev-Agent, an automated system for repository crawling, environment setup, call chain analysis, and LLM evaluation. We also proposed Codev-Bench, a developer-centric, real-world framework that assesses whether code completion tools capture developers' intent across diverse contexts. Our evaluation of several state-of-the-art LLMs offered insights for future improvements in code completion models.  Looking forward, we aim to extend Codev-Agent to support multiple programming languages, generating a multilingual dataset, while also refining Codev-Bench to handle more complex developer scenarios.

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

# Codev-Bench: How Do LLMs Understand Developer-Centric Code Completion?

# Appendices

## A  MORE EVALUATION RESULTS OF CODEV-BENCH

In contrast to Table 3, we present a more detailed evaluation of the results in Table 4, Table 5, Table 6 and Table 7. For each scenario, we assess the test pass rates corresponding to various types of code blocks, including functions, logical condition blocks, loop blocks, exception handling blocks, and ordinary statements. Furthermore, we compare the performance of general LLMs with code LLMs. The general LLMs utilize a prompt format based on natural language, while the code LLMs employ a fill-in-middle prompt format.

Table 4: Comparing Pass@1 of different models on full block completion subset.

| Model | Function | If | For | Try | Statement | Average |
|---|---|---|---|---|---|---|
| General LLMs (Natural Language Mode) | | | | | | |
| GPT-4 | 15.38 | 22.08 | 12.70 | 15.22 | 50.00 | 27.56 |
| GPT-4o | 23.08 | 42.86 | 33.33 | 26.09 | 69.00 | 45.19 |
| GPT-4o-mini | 0.00 | 16.88 | 14.29 | 2.17 | 37.00 | 19.23 |
| Claude-3.5-Sonnet | 0.00 | 7.79 | 6.35 | 0.00 | 46.00 | 17.95 |
| Deepseek-v2 | 0.00 | 1.30 | 0.00 | 0.00 | 9.00 | 3.21 |
| Mistral-123b | 0.00 | 3.90 | 0.00 | 0.00 | 24.00 | 8.65 |
| Yi-1.5-34b | 0.00 | 3.90 | 0.00 | 0.00 | 9.00 | 3.85 |
| Qwen-2-54b-moe | 0.00 | 1.30 | 0.00 | 0.00 | 6.00 | 2.24 |
| Qwen-2-72b | 0.00 | 19.48 | 4.76 | 0.00 | 69.00 | 27.88 |
| Llama-3.1-70b | 0.00 | 9.09 | 1.59 | 0.00 | 66.00 | 23.72 |
| Llama-3.1-405b | 0.00 | 23.38 | 28.57 | 15.22 | 78.00 | 38.78 |
| Code LLMs (Fill-In-Middle Mode) | | | | | | |
| Codeqwen-1.5 | 26.92 | 36.36 | 20.63 | 47.83 | 52.00 | 39.10 |
| Deepseek-coder-v2-lite | 53.85 | 32.47 | 33.33 | 56.52 | 61.00 | 47.12 |
| Codegeex-4-9b | 11.54 | 11.69 | 19.05 | 17.39 | 20.0 | 16.67 |
| Starcoder-2-7b | 0.00 | 2.60 | 0.00 | 2.17 | 1.00 | 1.28 |
| Codegemma-7b | 46.15 | 41.56 | 33.33 | 71.74 | 70.00 | 53.85 |

## B  THE CORRELATION BETWEEN PASS@1 AND ES

In this paper, we analyze the prediction results of various models across four scenarios, evaluating them based on test pass rates and edit distance similarity. Our goal in this section is to assess whether there is consistency between the predictions of the test pass rates and edit distance similarity. An introduction to the test pass rates and edit distance similarity is provided in Section 4.3.

In the four scenarios, we calculate the average Pass@1 and average Edit Similarity (ES) values predicted by each general LLMs and code LLMs. To visualize these results, we plot each model's outcomes in Figure 6 using a scatter plot, with the X-axis representing the average Pass@1 and the Y-axis representing the average ES value. Additionally, we calculate the Pearson correlation between the average Pass@1 and average Edit Similarity (ES) values predicted by each model in the four

Table 5: Comparing Pass@1 of different models on inner block completion subset.

| Model | Function | If | For | Try | Statement | Average |
|---|---|---|---|---|---|---|
| General LLMs (Natural Language Mode) | | | | | | |
| GPT-4 | 10.0 | 16.55 | 19.05 | 10.87 | 14.53 | 15.47 |
| GPT-4o | 2.5 | 13.67 | 19.05 | 0.0 | 11.17 | 12.08 |
| GPT-4o-mini | 2.5 | 3.6 | 2.38 | 4.35 | 2.79 | 3.02 |
| Claude-3.5-Sonnet | 37.5 | 46.76 | 46.83 | 50.0 | 54.19 | 48.87 |
| Deepseek-v2 | 20.0 | 30.94 | 19.84 | 32.61 | 22.35 | 24.72 |
| Mistral-123b | 7.5 | 4.32 | 7.14 | 4.35 | 1.12 | 4.15 |
| Yi-1.5-34b | 12.5 | 14.39 | 13.49 | 23.91 | 16.20 | 15.47 |
| Qwen-2-54b-moe | 35.0 | 23.02 | 20.63 | 41.3 | 18.99 | 23.58 |
| Qwen-2-72b | 15.0 | 8.63 | 7.14 | 6.52 | 7.82 | 8.30 |
| Llama-3.1-70b | 0.0 | 5.76 | 6.35 | 2.17 | 6.70 | 5.47 |
| Llama-3.1-405b | 15.0 | 23.74 | 34.13 | 8.70 | 16.76 | 21.89 |
| Code LLMs (Fill-In-Middle Mode) | | | | | | |
| Codeqwen-1.5 | 75.0 | 38.85 | 65.87 | 47.83 | 56.42 | 54.72 |
| Deepseek-coder-v2-lite | 66.67 | 68.48 | 87.21 | 100.0 | 79.82 | 78.62 |
| Codegeex-4-9b | 22.22 | 36.96 | 26.74 | 100.0 | 27.52 | 32.70 |
| Starcoder-2-7b | 0.0 | 0.0 | 0.0 | 0.0 | 0.0 | 0.0 |
| Codegemma-7b | 88.89 | 71.74 | 79.07 | 76.92 | 78.90 | 77.36 |

Table 6: Comparing Pass@1 of different models on incomplete suffix completion dataset.

| Model | Function | If | For | Try | Statement | Average |
|---|---|---|---|---|---|---|
| General LLMs (Natural Language Mode) | | | | | | |
| GPT-4 | 0.00 | 0.00 | 0.79 | 13.04 | 15.31 | 6.94 |
| GPT-4o | 0.00 | 1.95 | 1.59 | 8.70 | 10.20 | 5.32 |
| GPT-4o-mini | 0.00 | 1.30 | 0.79 | 5.43 | 4.08 | 2.58 |
| Claude-3.5-Sonnet | 1.92 | 3.25 | 7.14 | 6.52 | 13.78 | 7.74 |
| Deepseek-v2 | 0.00 | 0.65 | 0.00 | 2.17 | 2.55 | 1.29 |
| Mistral-123b | 0.00 | 0.00 | 0.00 | 0.00 | 1.53 | 0.48 |
| Yi-1.5-34b | 0.00 | 0.00 | 0.00 | 0.00 | 0.00 | 0.00 |
| Qwen-2-54b-moe | 0.00 | 0.00 | 0.00 | 0.00 | 0.51 | 0.16 |
| Qwen-2-72b | 0.00 | 0.00 | 2.38 | 2.17 | 4.08 | 2.10 |
| Llama-3.1-70b | 0.00 | 0.00 | 0.00 | 1.09 | 4.08 | 1.45 |
| Llama-3.1-405b | 0.00 | 1.30 | 0.79 | 6.52 | 16.84 | 6.77 |
| Code LLMs (Fill-In-Middle Mode) | | | | | | |
| Codeqwen-1.5 | 0.00 | 1.57 | 5.10 | 7.76 | 9.02 | 5.44 |
| Deepseek-coder-v2-lite | 0.00 | 0.94 | 3.49 | 10.42 | 5.08 | 3.75 |
| Codegeex-4-9b | 0.00 | 0.00 | 0.00 | 0.00 | 1.69 | 0.50 |
| Starcoder-2-7b | 0.00 | 0.00 | 0.00 | 0.00 | 0.85 | 0.25 |
| Codegemma-7b | 0.00 | 3.64 | 3.05 | 10.00 | 6.38 | 4.84 |

scenarios as follows,

$$r = \frac{\sum_{i=1}^{n}(P_i - \bar{P})(E_i - \bar{E})}{\sqrt{\sum_{i=1}^{n}(P_i - \bar{P})^2}\sqrt{\sum_{i=1}^{n}(E_i - \bar{E})^2}}, \tag{1}$$

where, $P_i$ refers to the Pass@1 value of $i$-th model and $E_i$ refers to the ES value of $i$-th model, $\bar{P}$ and $\bar{E}$ refer to the average value of each model's Pass@1 and ES value, $n$ refers to the number of models to be tested.

Table 7: Comparing Pass@1 of different models on RAG-based completion subset.

| Model | Function | If | For | Try | Statement | Average |
|---|---|---|---|---|---|---|
| General LLMs (Natural Language Mode) | | | | | | |
| GPT-4 | 11.54 | 18.18 | 12.70 | 10.87 | 49.00 | 25.32 |
| GPT-4o | 15.38 | 35.06 | 34.92 | 19.57 | 69.00 | 41.99 |
| GPT-4o-mini | 0.00 | 11.69 | 19.05 | 0.00 | 43.00 | 20.51 |
| Claude-3.5-Sonnet | 0.00 | 3.90 | 0.00 | 2.17 | 47.00 | 16.35 |
| Deepseek-v2 | 0.00 | 1.30 | 1.59 | 0.00 | 26.00 | 8.97 |
| Mistral-123b | 3.85 | 3.90 | 0.00 | 0.00 | 33.00 | 11.86 |
| Yi-1.5-34b | 0.00 | 1.30 | 0.00 | 0.00 | 7.00 | 2.56 |
| Qwen-2-54b-moe | 0.00 | 0.00 | 0.00 | 0.00 | 6.00 | 1.92 |
| Qwen-2-72b | 0.00 | 16.88 | 4.76 | 2.17 | 66.00 | 26.60 |
| Llama-3.1-70b | 0.00 | 7.79 | 1.59 | 0.00 | 71.00 | 25.00 |
| Llama-3.1-405b | 3.85 | 22.08 | 33.33 | 19.57 | 78.00 | 40.38 |
| Code LLMs (Fill-In-Middle Mode) | | | | | | |
| Codeqwen-1.5 | 42.31 | 31.17 | 30.16 | 41.30 | 54.00 | 40.71 |
| Deepseek-coder-v2-lite | 0.00 | 0.00 | 0.00 | 0.00 | 0.00 | 0.00 |
| Codegeex-4-9b | 7.69 | 10.39 | 17.46 | 23.91 | 23.0 | 17.63 |
| Starcoder-2-7b | 0.00 | 1.30 | 0.00 | 0.00 | 2.00 | 0.96 |
| Codegemma-7b | 42.31 | 45.45 | 44.44 | 65.22 | 70.00 | 55.77 |

Finally, the correlations in all the four scenarios are 0.793, 0.991, 0.529 and 0.822. This indicates that, in most cases, using edit similarity as a statistical measure does not objectively reflect the quality of the model's generation in the code completion task.

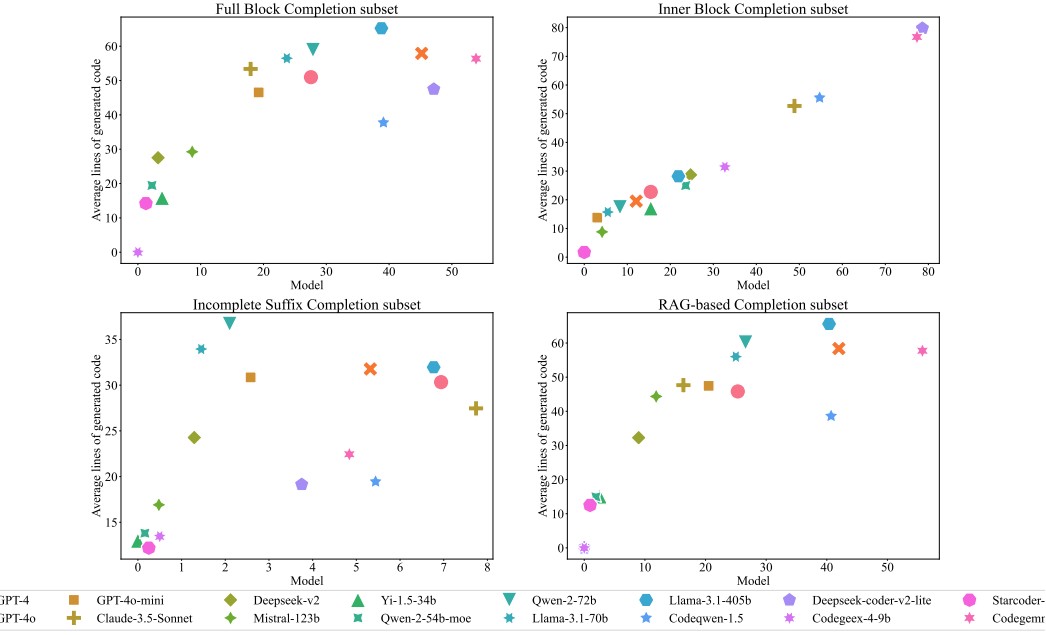

Figure 6: Comparing the correlation of average Pass@1 and average ES in different models.

# C Complete Prompt of General LLMs to Accomplish Code Completion Tasks

We designed prompts to enable general LLMs to comprehend both the preceding and following context of code, allowing them to predict the exact code snippets that fit between these contexts. This serves as an agent for the code completion task. The specific prompts are as follows:

```
If you were a code completion agent, I would provide you with a snippet of code, and you would
↪  need to return the completed code segment. The content after <filename> indicates the name
↪  of the file to complete, the content after <fim_prefix> indicates the content from the
↪  cursor position to the beginning of the file (the context), and the content after
↪  <fim_suffix> indicates the content from the cursor position to the end of the file (the
↪  continuation). You need to predict the content that should be completed between the
↪  context and the continuation. The predicted completion should be output after
↪  <fim_middle>.

# Example
Original Code:
```
<filename>solutions/solution_1.py<fim_prefix># Here is the correct implementation of the code
↪  exercise
def maxPresum ( a , b ) :
    """
    Maximum Prefix Sum possible by merging two given arrays
    """
    <fim_suffix><fim_middle>
```
Completion Needed:
```
X = max ( a [ 0 ] , 0 )
    for i in range ( 1 , len ( a ) ) :
        a [ i ] += a [ i - 1 ]
        X = max ( X , a [ i ] )
    Y = max ( b [ 0 ] , 0 )
    for i in range ( 1 , len ( b ) ) :
        b [ i ] += b [ i - 1 ]
        Y = max ( Y , b [ i ] )
    return X + Y
```
# Task
Original Code:
```
question here
```
# Important Points
## Point 1
- If the current code block that needs completion is a statement, just complete that statement
↪  without completing any subsequent statements.
- If the current code block that needs completion is a class/function, complete that
↪  class/function in its entirety.
- If the current code block that needs completion is a logical block (e.g. if, for, while, try,
↪  etc.), complete the entire current logical block.
- If the current code block is already complete, return nothing (an empty response).
## Point 2
Only return the recommended code snippet for completion; do not return the original code. The
↪  returned code should be enclosed in ``````.

"""
Completion results:
```
code for completion
```
"""
```

# D Major types of completion errors

In this section, we introduce several common types of code completion errors, including the inability to stop at the correct position, failure to recognize that completion is within one code block, and the inability to identify when no content should be completed.

## D.1 STRUGGLING TO STOP GENERATING CONTENT AT THE APPROPRIATE POSITION

In this example, the model predicts the target statement "correction_term_mean = np.mean(np.real(kernel_values, axis=-1))", however, it continues to predict some other code lines. This makes the unit test fail to pass. That is to say, this model encounters the error of struggling to stop generating content at the appropriate position.

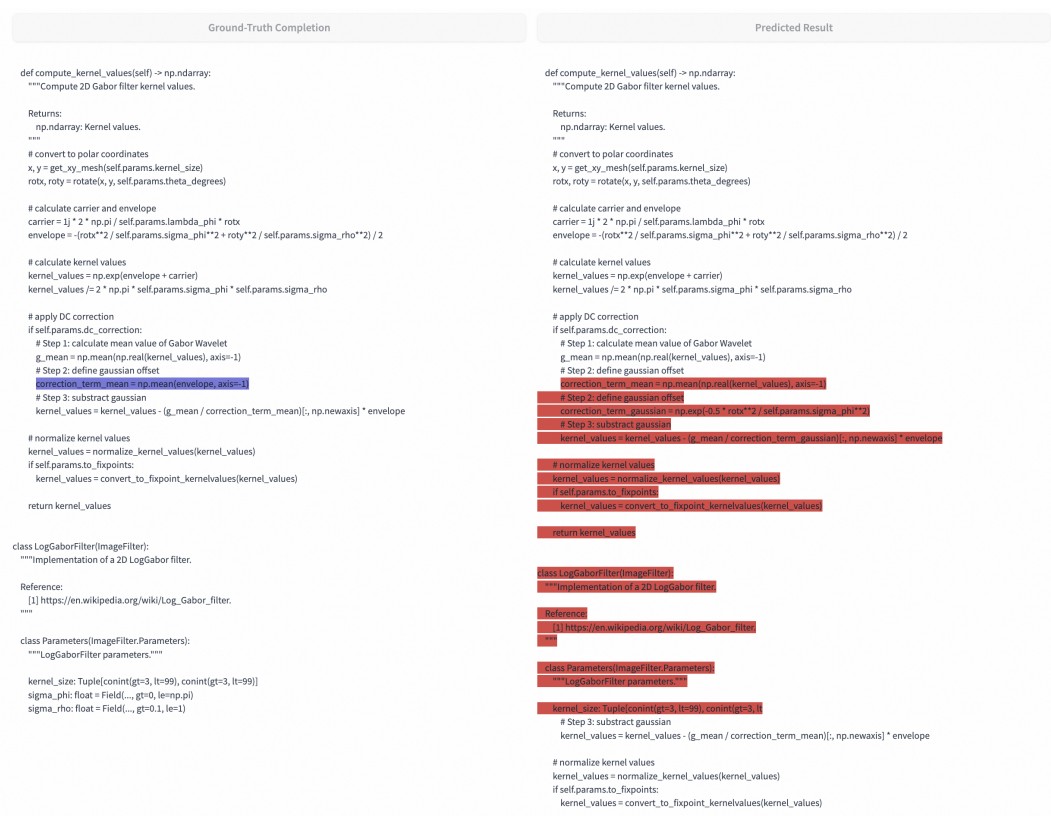

Figure 7: Completion error of struggling to stop generating content at the appropriate position.

## D.2 STRUGGLING TO RECOGNIZE THAT COMPLETION IS WITHIN ONE CODE BLOCK

In this example, the model only needs to predict " += " symbol to fill the code within one code block "self._id_count += len(instances)". However, this model predict " = len(self._prev_instances)" and then write another "else" block. This error indicates that the model fail to recognize that this type of completion is within one code block.

## D.3 STRUGGLING TO IDENTIFY WHEN NO CONTENT SHOULD BE COMPLETED

In this example, "mask = norms > min_distance_between_sector_points_in_px" is already completed, thus the model only need to predict nothing. However, this model still predicts two code lines. Although there are no syntax errors in this prediction, but this might cause significant disruption to users and makes the unit test fail to pass. This type of error indicates the model struggling to identify when no content should be completed.

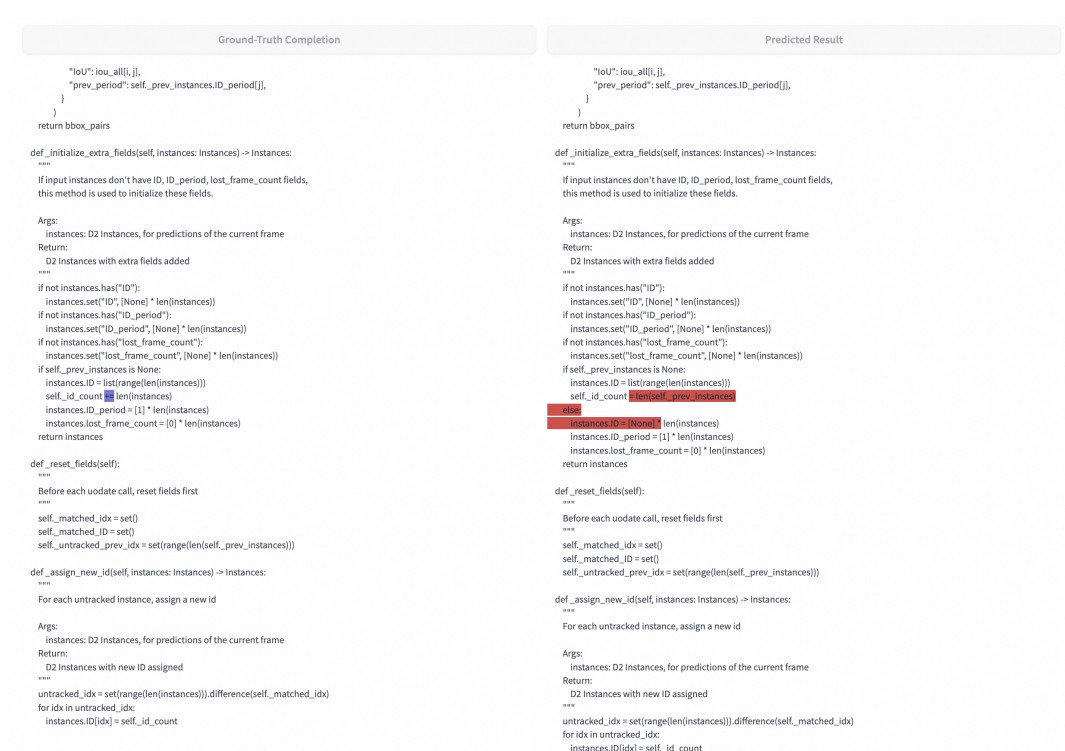

Figure 8: Completion error of struggling to recognize that completion is within one code block.

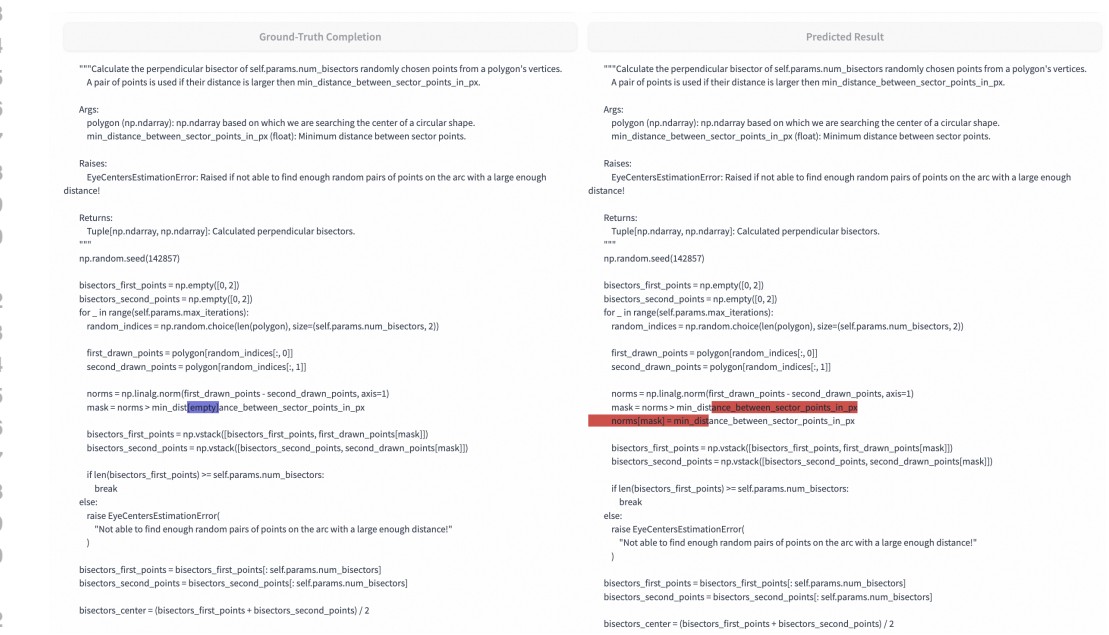

Figure 9: Completion error of struggling to identify when no content should be completed.

