# OpenReview forum: "Codev-Bench: How Do LLMs Understand Developer-Centric Code Completion?"
_ICLR.cc/2025/Conference — Submitted to ICLR 2025_

### Official Review · Reviewer_Mpaw · 2024-11-04

**Soundness:** 2
**Presentation:** 1
**Contribution:** 2
**Rating:** 3
**Confidence:** 5

**Summary:**

This paper introduces a new benchmark for evaluating code generation that both uses execution signal from real test cases and bases its sample collection on insights from real-world code completion usage. It produces a dataset of code completion contexts at various granularities in Python repositories and evaluates a wide range of LLMs on these. This shows that models perform relatively poorly on many prompts, with code-specific models often yielding the best scores.

**Strengths:**

This work provides an automated method for extracting executable, realistic code completion test cases from real-world code repositories (Codev-Agent), which is a valuable and complex tool. It also evaluates a wide range of LLMs, offering some useful insights across a range of settings.

**Weaknesses:**

The dataset curated in this work is very small and poorly reflects the stated motivation of learning from the industrial analysis in Sec. 3. The methodology has a number of problems. And the writing includes a number of invalid claims and frequently lacks clarity. These concerns are discussed in order below, followed by a number of minor issues.

The main artifact in this work is the dataset curated by Codev-Agent. This dataset spans just 10 projects, only in Python, across which the test cases cover just 55 files (Sec. 3.3.2). This is an insufficiently large and diverse dataset size to draw meaningful conclusions about model performance. To be clear, automatically generating executable test environments from real repositories is challenging, so it is understandable that the resulting dataset size is smaller than non-executable counterparts. But realistic evaluations need to involve substantially more distinct test cases, preferably spanning multiple programming languages and distinct domains.

On a related note, this limitation stands in sharp contrast to the strong claims made in Sec. 3.3.1, which should be toned down. It is unreasonable to argue that all other benchmarks listed here are not extensible and require manual effort to extend (that certainly does not apply to benchmarks like RepoBench, which scraped 25K repositories from GitHub, clearly not by hand), and the inclusion of an "Agent" column seems redundant with the extensibility one. It is especially improper to state that Codev-Bench's extensibility even applies to repositories with few to no tests in that it just requires writing some test cases (L314), which surely applies to all other benchmarks as well then. Similarly, the argument that RepoMasterEval does not "deep dive" enough into user needs to qualify seems unnecessary, especially given the concerns about the industrial analysis below.

Secondly, the connection between the dataset, at least in terms of the scenarios discussed in Sec. 4, and the industrial/business analysis in Sec. 3 is not clear. The latter analysis mainly shows that completions tend to involve non-control-flow statements, require completing an entire single line, and come with a wide variety of contexts. The actual evaluation considers completions at the block level, where functions, control-flow, and other tatements are lopped in together. Although the appendix subsequently breaks down performance by statement type, it is not clear where the observation regarding completion types factors in. The analysis in this work does not reweight aggregate scores by completion type, or otherwise adjust its conclusions based on Fig. 1.a's distribution. The observation that most completions concern a single line is not utilized at all in the rest of the paper. Nor is the idea that many completions come with just a few tokens of context (itself highly surprising in real-world programming, see the concern below).

On that note, it is quite concerning that the origin and scale of this dataset is not provided so we can evaluate whether it is, in fact, representative of realistic code completion usage. The paper does not discuss what LLM (or other model) is used as a code completion model in the original data, nor under what constraints it operates (e.g. when is it triggered, how well it handles single vs. multi-line completions). Many other details are missing too, such as what exactly a "general statement" is. It would be better to use established nomenclature from program parsers. Similarly, in Fig. 1.B, does "entire lines" mean multiple lines? And why are so many completions "empty"? Completions are normally triggered by edit actions, in which case an empty completion is never correct. In Fig. 1.c, is the "prompt" the entire file-level context at the time the completion is invoked? If so, why are so many contexts (nearly) empty? Completing the first token in a file must be a very rare occurrence. These insights also all seem quite far from "fine-grained". Many empirical studies have been conducted on general properties of code completion queries in relation to model performance. I would expect a fine-grained study to offer more specific insights.

Finally, the prompting setup raises a number of concerns, which are reflected by the remarkably poor results of most models on most tasks. In particular:
- The prompt shown in App. C makes it quite unclear how to leverage the suffix. It only shows a single example, which contains an empty suffix. The prompt should at least include sample-appropriate demonstrations for the different types of scenarios. It may also be noted that using the language of FiM is not necessarily suitable for post-trained models. A more natural format would show the complete code file with a cursor placeholder. This may well be the cause of the pattern the paper notes in which the models continue their completion beyond the target line(s). The example shown in Fig. 7 (App. D.1.) strongly suggests that the model has failed to understand the task, as it copies the input code verbatim after generating the missing line. That points at a prompting failure.
- The prompt also repeatedly tells the model to provide a (non-empty) completion, only mentioning the possibility of an empty response towards the end. That may well be the reason for models often failing to generate empty completions.
- There are a number of other odd statements in the prompt, such as the start ("If you were a code completion agent" instead of a concrete instruction) and the end (which seems to tell the model how to format the output twice, in two different ways).

All this probably place a big role in the remarkably poor performance reported for most models and tasks in Tab. 3 - 7. These numbers are far below values on other benchmarks, which strongly suggests the models are being queried with insufficient context and/or misleading prompts (and/or the evaluation harness has serious issues).


Minor Issues:

- The citation style appears to be incorrect.
- L12: "most frequent" -> "most frequently used"?
- L34: multiple typos
- The first three pages repeat the challenges from prior work (that motivate this paper) rather often (e.g. "coarse-grained tasks" is mentioned five times); consider reducing this redundancy.
- L250: how exactly does "fusing" work here?
- L287/288: "by by" -> "by one"
- Tab. 2: this table is entirely unclear. Do "Scene1-4" refer to scenarios? If so, those are not introduced as concepts until the next section, and conflict with the title, which says "Different projects" (of which there should be 10). What are these averages of?
- L509: ends abrumptly, missing text.

**Questions:**

Data:
- Given the small sample size, did you conduct statistical significance tests on the results? Please report the typical confidence intervals.
- Are the 10 Python projects considered in this study from reasonably diverse ecosystems? How even was the distribution of code files exercised per project? This is important because there were just 55 of the former over 10 projects total.
- Please aim to tone down the claims in Sec. 3.3.1.
- What are some concrete ways in which the industrial analysis relates to the benchmark construction and results?
- Did you extract any more fine-grained insights from the industrial analysis?
- Why are so many of the prompts empty or nearly empty in Fig. 1.c? Please also discuss all the other questions around the specifics of the tool from which the data was obtained.

Methods:
- Please improve the clarity of the prompt to better identify which failures are due to model performance vs. prompt clarity.
- Did you conduct any systematic manual evaluation of the generated completions with high failure rates to ensure that there are no issues in the execution harness?
- What explains the very low correctness rates in most of the results, aside from (potentially) prompting issues making it difficult for the models to know how much code to complete? Are the models provided with sufficient context to infer the missing code (e.g. would a human be able to)?

---

> ### Author Response · Authors · 2024-12-02
>
> Dear Reviewer,
>
> We greatly appreciate the time and effort you have taken to review our work. Your feedback is extremely valuable to us and will help us improve the quality of our paper.
>
> In our next steps, we plan to further expand the dataset and make substantial revisions to enhance the overall quality of the manuscript. We will also include more comprehensive details to address your comments and ensure a clearer presentation of our work.
>
> Thank you once again for your thoughtful and constructive review.
>
> Best regards,
>
> Authors

---

### Official Review · Reviewer_3ADN · 2024-11-04

**Soundness:** 3
**Presentation:** 2
**Contribution:** 3
**Rating:** 5
**Confidence:** 4

**Summary:**

This paper proposes CodevBench, a repository-level benchmark to evaluate a large language model's skill in real-world, developer-centric code completion task. To streamline the benchmark creation, authors introduce CodevAgent, an LLM-based agent that automates the collection of the latest Github repositories, the execution environment setup, the analysis of call chains and the generation of test samples. Authors also evaluate various General LLMs and Code LLMs to complete the tasks.

**Strengths:**

- The authors proposed CodevAgent as a framework to automatically generate new test samples and evaluate LLMs without the need for extensive manual annotations. This can be crucial for preventing data leakage and for effectively scaling the benchmark.
- The idea of collecting and analyzing true needs of software developers could be quite impactful, given the relevance to downstream applications.
- The authors conducted a comprehensive evaluation of multiple General LLMs and Code LLMs.
- The evaluation metric 'Edit Similarity' is a helpful criterion for evaluating code quality, particularly for fairly judging some minor errors that can be easily corrected.

**Weaknesses:**

- The details regarding the construction of the benchmark with CodevAgent are somewhat unclear. For instance, while CodevAgent appears to utilize unit tests from the raw repositories for evaluation, the authors do not clearly explain how to ensure the quality of these test cases or the selection of benchmark repositories, given that raw repository tests might not be reliable. It is also unclear the selection rate, data quality, and time investment in the transition from raw repositories to well-generated test samples via CodeAgent.

- The analysis presented in this paper is somewhat high-level. While the authors provide examples of error cases, the detailed input is missing, and the presented examples rely heavily on specific context. A statistical analysis of the highlighted error cases would enhance the robustness of the findings.

- In section 3.1 Product Business Data Analysis, the authors present data analysis results to illustrate the needs of expert developers. However, there is no explicit description of how the source data was acquired, who was interviewed, or the scale of these interviews.

- There are instances of repeated sentences, such as:

  - "Automates repository crawling, constructs execution environments, extracts dynamic call..." appears in both the abstract (Line 023), introduction (Line 061) and methodology (Line 150).

  - "three common challenges: (1) ..." appears in both the 2.3 Benchmark For Code Completion (Line 136) and the Methodology (Line 150).

  To enhance clarity, it would be beneficial to either remove duplicate content or revise the phrasing.

- There are some presentation issues:

  - Line 509: We also show some. It seem to lack some additional words.

  - Figure 3 is not cited within the main body.

  - Table 2 is mentioned but lacks an explanation of the significance of the numbers presented.

  - Tables 6 and 7 are included in the appendix, while the analyses appear in the main body. It would be more coherent to move the tables and the corresponding analyses together.

  - The figures related to completion errors are not referenced in the relevant analysis sections of the appendix.

**Questions:**

- How does CodevAgent ensure successful crawling and environment setup? Given the complexity and variability of raw repositories, this is a time-consuming task even for humans.

- Can the authors further explain how the "Edit Similarity" metric is applied within CodevBench? It seems to be no direct results reported in the main body related to this metric.

- How is the distribution of test cases managed across the ten repositories in CodevBench? Have the authors considered the differences in complexity and domains among various repositories? When adding new test samples generated by CodevAgent, how do the authors balance the difficulty differences with existing repository test samples?

- The authors suggest that CodevBench supports user customization and optimization. Could the authors clarify how this functionality works? The paper seems to lack explicit explanation in this regard.

---

> ### Author Response · Authors · 2024-12-02
>
> Dear Reviewer,
>
> We are deeply grateful for your time and effort in reviewing our work. Your insightful comments are invaluable to us.
>
> In our revision, we will work on further expanding the dataset and refining the manuscript to enhance its quality. We will also provide additional details to ensure greater clarity and address any concerns raised during the review process.
>
> Thank you again for your thoughtful feedback, which will help us significantly improve this work.
>
> Best regards,
>
> Authors

---

### Official Review · Reviewer_UsaN · 2024-11-04

**Soundness:** 3
**Presentation:** 4
**Contribution:** 3
**Rating:** 6
**Confidence:** 4

**Summary:**

This paper addresses the problem of repository-level code completion, while focusing on a wide variety of granularity, including completing logic blocks, individual statements, filling in comments, etc. Such a design departs from prior work, which are specifically focused on only one of line-level, function-level, or class-level completion; and is grounded in an empirical study from analyzing industry data. The authors adopt an agent-based approach to automate the data collection process, *minimizing manual efforts* in this regard. By comparing multiple state-of-the-art LLMs, both general-purpose and code-specific, the authors aim to redefine evaluation criteria for code completion tools by integrating real-world developer feedback and contextual variability.

**Strengths:**

- *Data-driven design*: The benchmark is grounded in insights from an industrial code completion tool, allowing Codev-Bench to closely align with real-world developer workflows and capture diverse usage scenarios. This industry-driven design enhances the benchmark's practical significance.
- Moves beyond traditional function-level tasks by introducing scenario-based evaluation with granular tasks.
- Extensive experimental results across four different completion scenarios, showcasing the strengths and limitations of both general and code-specific LLMs

**Weaknesses:**

- *Limited discussion/details on the evaluation of completed code at different levels of granularity*: For in-line or single-line code completion, testing against all unit tests might be excessive. Since these completions are smaller and more context-specific, they often don’t impact the broader program behavior significantly.
- *Benchmark adaptability*: The focus on industrial-level repositories and high-quality data is valuable, yet the paper could discuss more on the adaptability of Codev-Bench to repositories with minimal unit testing or incomplete documentation, which are common in many open-source settings. Addressing these would improve the applicability of Codev-Bench in diverse dev environments.
- *Agent v/s manual data collection*: The authors argue an agent-based data collection process helps with making the benchmark more extensible to new repositories/programming languages. While that's indeed the case, a manual data collection is also often focused on code quality.

**Questions:**

1. Can you give more details on the unit test filtering for an instance with different granularity levels?
2. How adaptable is Codev-Bench to repositories with limited or poorly structured unit tests, as is common in many open-source projects? This affects the practical usefulness of this benchmark. Can you give more details in this regard?
3. The authors emphasize the extensibility benefits of the agent-based data collection process. Have you evaluated the potential trade-offs in data quality or representativeness compared to manual collection methods, especially regarding the nuances in code quality and repository structure?

---

> ### Author Response · Authors · 2024-12-02
>
> Dear Reviewer,
>
> Thank you very much for taking the time to review our manuscript. We sincerely appreciate your constructive feedback and valuable suggestions.
>
> We plan to expand the dataset further and will revise the manuscript to improve its overall quality. In our revision, we will also include additional details to address the points raised and ensure a more comprehensive and robust presentation of our work.
>
> Thank you again for your time and effort in reviewing our submission.
>
> Best regards,
>
> Authors

---

### Official Review · Reviewer_e3hN · 2024-11-04

**Soundness:** 2
**Presentation:** 2
**Contribution:** 2
**Rating:** 3
**Confidence:** 4

**Summary:**

This paper produces the Code-Development Benchmark Codev-Bench, and the Codev-Agent system that automates repository crawling

Section 2 summarizes related work in the areas of the use of LLMs for code generation, benchmarking code generation, and benchmarking code completion.

Section 3 starts by describing code completion experience in an industrial setting, covering (1) the trigger points for code completion (almost 70% on statements); (2) the number of lines completed (almost 50% is full line); and (3) the length of the user prompts (41% contain more than one token).

It then describes Codev-Agent,, which crawls repositories, sets up an execution environment, conducts static and dynamic call chain analysis, and generates unit tests.
It then uses a fill in the blank approach to compare various language models, measuring their effectivness in terms of edit distance and test pass rate.

Subsequently, Codev-Agent is used to create Codev-Bench. This currently consists of 862 (Python) code files and 191 test files extracted from 10 repositories.

Codev-Bench is subsequently used in four scenarios (full block, inner block, incomplete suffix, and RAG-based completion) with state of the art general and code-specific LLMs.

Insight reported from this experiment include:

1. Simpler block-like statements are easier to complete
2. Code LLMs outperform general LLMs in block completion tasks
3. Both general LLMs and code LLMs fail in incomplete suffix completions
4. Also in a RAG setting, code LLMs outperform general LLMs for coding tasks

**Strengths:**

- Nice to see a benchmark inspired by actual use
- Neat trick to generate test cases to assess code completion quality

**Weaknesses:**

The 'product business data analysis' description lacks detail. How many data points were collected? From what type of people? Using which language models? Was this a standard product like GitHub co-pilot or something else? How was line completion integrated in the IDE? There are also several existing publications reporting on industrial or practical usage of line completion, using different categories. See, e.g., https://arxiv.org/abs/2402.16197

Also for Codedev-Bench, crucial details are missing, such as the actual 10 repositories used, the actual files, test cases, etc.

It is unclear to me what the dynamic and static call chain analysis actually does, and how it informs test case generation. The process described is reminiscant of mutation testing, where the code under test is mutated to explore which tests will fail.
Since tests are used, it is crucial to report the quality (adequacy) of the tests in place, for example in terms of branch coverage.

The paper states:

> we systematically extract test samples that match the distribution of real-world business scenarios

This makes sense, but the paper doesn't show any example or any detail that allows the reader to assess this claim, nor to reproduce this result. What were concrete 'real-world scenarios' for the 10 repositories chosen? How many of such scenarios were identified?

The paper has no discussion section offering an interpretation of the results, or an asssessment of the results' limitations.

The completion setting generated is still fairly artificial, as it is essentially fill in the blank. This assumes a correct context exists, in which just a few lines need to be added. In actual software development, the code will be incomplete, and most likely changes across different file locations or files are needed. This is not discussed.

Ultimately, the four findings are unsurprising, as they primarily indicate that code LLMs are better at coding tasks than general LLMs.

While Codedev-Agent seems impressive machinery, the resulting benchmark seems small, with just 296 code blocks. Why is this?

Lastly: were the repositories chosen part of the training data for the language models selected? Please discuss.

**Questions:**

- How good are the tests provided
- Which 10 repositories did you investigate
- What was the industry context of the business data analysis, and how much data was collected in what way?

---

> ### Author Response · Authors · 2024-12-02
>
> Dear Reviewer,
>
> Thank you for dedicating your time to review our paper and for providing us with valuable feedback. We truly appreciate your thoughtful insights.
>
> In the upcoming revision, we will focus on expanding the dataset and revising the manuscript to improve its overall quality. Additionally, we will include more detailed explanations to address your feedback comprehensively and ensure the clarity of our work.
>
> Once again, we sincerely thank you for your constructive comments, which will greatly contribute to the improvement of our paper.
>
> Best regards,
>
> Authors

---

### Meta-Review · Area_Chair_wufo · 2024-12-20

**Metareview:**

The paper introduces Codev-Bench, a benchmark for evaluating code completion at a repository level, aiming to reflect real-world developer workflows. To facilitate its creation, the authors developed Codev-Agent, an automated system for crawling repositories, setting up execution environments, performing call chain analysis, and generating test samples. The benchmark includes diverse code completion scenarios (full block, inner block, incomplete suffix, RAG-based). The core findings suggest that code-specific LLMs generally outperform general-purpose LLMs in code completion tasks, particularly in block completion and RAG settings. Simpler block-like statements are easier to complete. However, both types of LLMs struggle with incomplete suffix completions. The benchmark is intended to be extensible and reflect the distribution of real-world code completion needs based on industrial data analysis.

The main contribution of this paper is to create a benchmark grounded in real-world usage, as indicated by the industrial data analysis. The concept of automating test case generation with Codev-Agent is seen as a valuable contribution. The benchmark attempts to move beyond traditional function-level evaluations by incorporating various granularities of code completion tasks. The evaluation of multiple state-of-the-art LLMs across different scenarios provides initial insights.

The most significant weakness of this paper is the small size and limited diversity of the Codev-Bench dataset, spanning only 10 Python repositories and a small number of files. This raises concerns about the generalizability of the findings. The presentation is lack of clarity, including missing crucial details about the repositories and test cases, unclear explanations of Codev-Agent's functionality (static/dynamic analysis), and presentation issues (repeated sentences, uncited figures, unclear tables). The evaluation of code completion at different granularities is not well-justified, and the quality of the unit tests used for evaluation is a concern. Also, this paper lacks a thorough discussion section to interpret the results and acknowledge limitations.

Since all reviewers raised concerns about the dataset size and lack of detail regarding its creation and the industrial data analysis, I would recommend to reject this paper.

**Additional Comments On Reviewer Discussion:**

All reviewers raised concerns about the dataset size and lack of detail regarding its creation and the industrial data analysis. During rebuttal, the authors' responses were generally brief, stating their intention to expand the dataset and revise the manuscript for clarity. However, they did not provide specific answers to the reviewers' detailed questions or offer concrete solutions to the identified problems within the rebuttal.

---

### Decision · Program_Chairs · 2025-01-22

Reject